# Bevacizumab Augments the Antitumor Efficacy of Infigratinib in Hepatocellular Carcinoma

**DOI:** 10.3390/ijms21249405

**Published:** 2020-12-10

**Authors:** Thi Bich Uyen Le, Thanh Chung Vu, Rebecca Zhi Wen Ho, Aldo Prawira, Lingzhi Wang, Boon Cher Goh, Hung Huynh

**Affiliations:** 1Department of Haematology-Oncology, National University Health System, Singapore 119074, Singapore; thi_bich_uyen_le@nuhs.edu.sg (T.B.U.L.); csiwl@nus.edu.sg (L.W.); phcgbc@nus.edu.sg (B.C.G.); 2Laboratory of Molecular Endocrinology, Division of Molecular and Cellular Research, National Cancer Centre, Singapore 169610, Singapore; vu.thanh.chung@nccs.com.sg (T.C.V.); rebecca.ho.z.w@nccs.com.sg (R.Z.W.H.); aldo.prawira@nccs.com.sg (A.P.); 3Cancer Science Institute of Singapore, National University Hospital, Singapore 117599, Singapore

**Keywords:** hepatocellular carcinoma, pan-FGFR inhibitor, angiogenesis inhibitor, invasion, vessel normalization, hypoxia, growth inhibition

## Abstract

The fibroblast growth factor (FGF) signaling cascade is one of the key signaling pathways in hepatocellular carcinoma (HCC). FGF has been shown to augment vascular endothelial growth factor (VEGF)-mediated HCC development and angiogenesis, as well as to potentially lead to resistance to VEGF/VEGF receptor (VEGFR)-targeted agents. Thus, novel agents targeting FGF/FGF receptor (FGFR) signaling may enhance and/or overcome de novo or acquired resistance to VEGF-targeted agents in HCC. Mice bearing high- and low-FGFR tumors were treated with Infigratinib (i.e., a pan-FGFR kinase inhibitor) and/or Bevacizumab (i.e., an angiogenesis inhibitor). The antitumor activity of both agents was assessed individually or in combination. Tumor vasculature, intratumoral hypoxia, and downstream targets of FGFR signaling pathways were also investigated. Infigratinib, when combined with Bevacizumab, exerted a synergistic inhibitory effect on tumor growth, invasion, and lung metastasis, and it significantly improved the overall survival of mice bearing FGFR-dependent HCC. Infigratinib/Bevacizumab promoted apoptosis, inhibited cell proliferation concomitant with upregulation of p27, and reduction in the expression of FGFR2-4, p-FRS-2, p-ERK1/2, p-p70S6K/4EBP1, Cdc25C, survivin, p-Cdc2, and p-Rb. Combining Infigratinib/Bevacizumab may provide therapeutic benefits for a subpopulation of HCC patients with FGFR-dependent tumors. A high level of FGFR-2/3 may serve as a potential biomarker for patient selection to Infigratinib/Bevacizumab.

## 1. Introduction

Despite several treatment options for patients with early diagnosis, hepatocellular carcinoma (HCC) remains high in mortality rate, representing the third leading cause of cancer-related death in the world [1,2]. Surgical resection and liver transplant accounts for only a small number of early-stage HCC [3]. Due to a lack of visible symptoms in the early stage, as well as the rapid growth of the tumor, more than 80% of HCC cases are diagnosed in later stages [4]. In unresectable advanced diseases, poor prognosis with a one-year survival rate of less than 50% has been indicated [5]. Four multikinase inhibitors (Sorafenib, Regorafenib, Lenvatinib, and Cabozantinib) have been approved by the FDA for HCC treatment [6,7,8]. Clinical benefits of Sorafenib [6,7] and Lenvatinib [8] are at best modest and transient in improving the overall survival (OS) of HCC patients. Immune checkpoint inhibitors indicated for HCC confer modest activity and a randomized phase III study that evaluated Nivolumab versus Sorafenib as a first-line treatment in HCC patients (NCT02576509) did not meet its primary endpoint of OS. Therefore, novel therapies that effectively combat this deadly disease have yet to be discovered.

Previous studies suggested that growth factors and their corresponding receptors are commonly overexpressed and/or dysregulated in HCC. For example, aberrant overexpression/activation of the fibroblast growth factor (FGF)/fibroblast growth factor receptor (FGFR) pathways [9,10,11] and its activation have shown to contribute in HCC pathogenesis [10,12]. Overexpression of FGFR-2 and FGFR-3 are associated with tumorigenesis, metastasis, and poor prognosis of advanced HCC [13,14,15]. Furthermore, high expression of FGFR-2 in HCC has been correlated with multiple tumor nodules, distant recurrence, less tumor differentiation, portal vein invasion, a high level of alpha-fetal protein, and poor prognosis [14]. FGFs are potent angiogenic factors for hepatoma [16,17], and high expression of basic FGF has been detected in patients with HCC [18], where its activation via binding to FGFR led to downstream activation of several pathways including mitogen-activated protein kinase (MAPK) pathway and phosphoinositide-3 kinase-Akt-mTOR pathway [19]. These pathways regulate a number of cellular processes including oncogenesis, cell proliferation, migration, chemotaxis, differentiation, morphogenesis, and neo-angiogenesis [20,21,22]. In HCC, inhibition of MAPK led to a further increase in proapoptotic Bim and apoptosis and profound inhibition of cell proliferation [23].

Our preclinical study showed that Infigratinib had a high potency for FGFR-1, -2 and -3, but much lower potency for FGFR-4 [24,25]. Moreover, Infigratinib selectively inhibited the growth of FGFR-2/3-dependent HCC [23]. Likewise, FGFR signaling and its downstream targets, cell proliferation, angiogenic rescue program, hypoxia, invasion, and metastasis were inhibited. Infigratinib also induced vascular normalization and acted in synergy with Vinorelbine to promote apoptosis, suppress tumor growth, and improve the OS of mice compared to monotherapies [23]. The above observations suggest that targeting the FGFR pathway is a promising therapeutic strategy [21]. In fact, a number of clinical trials using FGF/FGFR pathway inhibitors have since been conducted or are ongoing in HCC.

The most prominent clinical features of HCC are hypervascularity, portal and hepatic invasion, and early metastasis. Regulation of HCC angiogenesis involves the presence of major factors such as vascular endothelial growth factor (VEGF), platelet-derived growth factor (PDGF), transforming growth factor (TGF)-α, TGF-β, FGFs, and hepatocyte growth factor (HGF) [26], where the complex interactions between these growth factors and cytokines result in the ultimate outcome of tumor vessel growth [27]. FGF levels increase after anti-VEGF treatment and modulate resistance to VEGF inhibition [28,29]. The combination of VEGF and basic FGF has been reported to have potent synergistic effects on HCC development and neovascular formation [30,31]. Given that VEGF and FGFR expression is correlated with angiogenesis, proliferation, and metastatic potential of tumor cells [23,26], inhibitors of the FGFR/VEGF receptor (VEGFR) signaling pathways may be an effective novel therapeutic intervention for HCC.

The goal of the present study is to achieve a better understanding of the mechanism underlying the anti-tumor effect of concurrent inhibition of VEGFR and FGFR1-3. We first examined the effect of Infigratinib combined with Bevacizumab on tumor angiogenesis, tumor cell invasion, metastasis, tumor hypoxia, and OS of mice bearing orthotopic HCC xenografts. Subsequently, we determined whether the inhibition of tumor growth was caused by reduced tumor cell proliferation, increase in apoptosis, or both. Here, we report the effects of combination therapy of Infigratinib/Bevacizumab on the carcinogenesis of HCC and OS in human HCC patient-derived xenograft (PDX) models that have been previously established and characterized for evaluation of new therapeutic agents for HCC [32].

## 2. Results

### 2.1. Infigratinib/Bevacizumab Demonstrates a Synergistic Effect that Inhibits Tumor Growth in High FGFR-Expressing HCC PDX Models

Since previous research showed an increase in newly capillary-like blood vessels with Infigratinib treatment [23], we sought to investigate if inhibiting VEGF activity would have additive or synergistic effects on the antitumor activity of Infigratinib. Mice bearing high and low FGFR-expressing tumors were treated with vehicle, Infigratinib, Bevacizumab, or Infigratinib/Bevacizumab for 8 to 20 days depending on the growth rate of each PDX. A dose of 5 mg/kg Bevacizumab weekly and 20 mg/kg Infigratinib daily significantly inhibited tumor growth for high FGFR-expressing HCC2006, HCC01-0909, HCC26-0808A, and HCC17-0211 (Figure 1a, *p* < 0.05), and, to a lesser extent, HCC13-0212 models (Figure 1b, *p* < 0.05). While Infigratinib had better antitumor activity than Bevacizumab in the tested models (Figure 1b), the combination treatment using Infigratinib/Bevacizumab resulted in superior inhibition of tumor growth than single agents, as seen in ectopic (Figure 1a,b, *p* < 0.05). Infigratinib had no significant antitumor activity against the HCC07-0409 model, which expresses low levels of FGFR2/3. The addition of Bevacizumab to Infigratinib did not augment the antitumor activity of Bevacizumab (Figure 1b and Appendix A). Our data indicate that inhibition of angiogenesis by Bevacizumab potentiated the antitumor activity of FGFR inhibition in FGFR-dependent HCC (Table 1). Throughout the course of treatment, no significant body weight loss or other signs of toxicity were observed in mice from treatment groups compared with those from the vehicle group (*p* = 0.67841), suggesting that the treatments are well-tolerated.

To determine if the addition of Bevacizumab to Infigratinib would increase severe adverse effects, we examined liver and kidney toxicities following Infigratinib/Bevacizumab treatments. As shown in Appendix A, daily treatment of mice with Infigratinib resulted in significant elevation in ALT, ALP, AST and 30% decrease in serum creatinine, suggesting some liver and kidney toxicities. These observations are consistent with the safety profiles of Infigratinib in human studies [33,34]. However, the addition of Bevacizumab did not cause further elevation of serum ALT, ALP, and AST as compared to Infigratinib monotherapy, suggesting that the combined Infigratinib/Bevacizumab did not cause more toxicity.

Since the FGFR pathway regulates a number of cellular processes including cell proliferation, migration, and neo-angiogenesis [21,22,23], the effects of Infigratinib and Infigratinib/Bevacizumab on blood vessel normalization, tumor hypoxia, tumor proliferation, and apoptosis were examined. HCC13-0109 and HCC06-0606Sor46 models were chosen for this study because they expressed high FGFR2/3 and responded well to Infigratinib and Infigratinib/Bevacizumab. Compared with the vehicle and Bevacizumab-treated groups, tumors harvested from Infigratinib-treated and combination-treated groups displayed significantly lower proportion of p-histone H3 Ser10 positive cells (Figure 2a, *p* < 0.05; Appendix A), suggesting that Infigratinib and Infigratinib/Bevacizumab inhibited cell proliferation. Combination-treated tumors showed a reduction in capillary-like blood vessels compared to Infigratinib-treated tumors by approximately 60%, suggesting that some capillary-like blood vessels were less sensitive to VEGF deprivation. Thus, our results were consistent with previous studies that showed how Infigratinib did not impair VEGF-induced blood vessel formations [24]. Although the amount of blood vessels in Bevacizumab/Infigratinib-treated tumors significantly reduced (*p* < 0.05), the surviving blood vessels were still functional as determined by lectin perfusion (Figure 2a and Appendix A). The combination significantly increased the number of cleaved PARP-positive compared to Infigratinib alone, suggesting that Bevacizumab enhanced the apoptotic activity of Infigratinib (Figure 2a and Appendix A, *p* < 0.05). Extensive hypoxia was observed in Bevacizumab-treated tumors presumably due to their antiangiogenic effects (Figure 2a). Despite a significant reduction in the number of functional blood vessels, the combination group showed either no change (HCC13-0109) or only a modest increase (HCC06-0606Sora46) in hypoxia when compared to the Infigratinib-treated group, indicating that the regions were well-oxygenated and the surviving blood vessels were productive. Quantification of mean microvessel density, lectin-positive blood vessels, p-Histone H3 Ser10, and cleaved PARP-positive cells are shown in Figure 2b.

Two HCC models’ different expression of FGF19 and FGFRs were used to determine the expression levels of regulators of angiogenesis such as PDGF-AA, CYR61, VEGF, TGF- β1, FGFs, and HGF [26] following treatments. HCC06-0606 expresses high levels of FGFR-2, -3, and -4 but undetectable FGF19. HCC13-0212 expresses low levels of FGFR2, modest levels of FGFR3, high levels of FGFR4 and high levels of FGF19. While the levels of CYR61 in Bevacizumab-treated tumors significantly increased, the levels of VEGF and HIF-1α mRNA in Infigratinib-treated HCC06-0606 tumors showed a slight decrease. Further reduction in the levels of VEGF, PDGF-AA, and CYR61 mRNA was observed in Infigratinib/Bevacizumab-treated tumors (Figure 2c and Appendix A). The levels of HGF and TGF-β1 were not altered by Infigratinib or Infigratinib/Bevacizumab treatments (Figure 2c and Appendix A). In the HCC13-0212 model, Infigratinib significantly inhibited the expression of VEGF, CYR61, and bFGF. The differences in the expression of FGF19 and FGFRs may be in part responsible for the observed differential effects of Infigratinib on the expression of angiogenic factors between the HCC13-0212 and HCC06-0606 models.

We next selected three high FGFR2/3 expressing models that showed strong response to Infigratinib/Bevacizumab (HCC13-0109, HCC06-0606 and HCC26-0808A) to study the effects of combined Infigratinib/Bevacizumab on the FGFR signaling pathway and its downstream targets. As shown in Figure 3, treatment of mice with Infigratinib, but not Bevacizumab, significantly decreased the expression of FGFR2-4 and p-FRS2-α, suggesting that the FGFR pathway was inactivated. Because the ERK and AKT/mTOR pathways are among the most critical cellular signaling pathways involved in hepatocarcinogenesis [35,36], the levels of p-ERK and mTOR pathways (p-p70S6K, p-S6R, and p-4EBP1) were examined. As shown in Figure 3, the levels of p-ERK1/2, p-p70S6K, p-S6R, and p-4EBP1 were significantly reduced, suggesting that ERK and mTOR pathways were inhibited (*p* < 0.05). Since positive cell cycle regulators play an important role in cell cycle progression and cell proliferation, the expression of survivin, cyclin B1, p-Cdc2, and p-Rb were investigated. Figure 3 shows that the expression of survivin, cyclin B1, p-Cdc2, and p-Rb were significantly downregulated (Figure 3, *p* < 0.05) by Infigratinib treatment. Treatment with Infigratinib/Bevacizumab led to a further reduction in the levels of FGFR2-4, survivin, cyclin B1, p-4EBP1, p-Cdc2, p-Rb, and Cdc25C. Significant elevation of cyclin dependent kinase inhibitor p27 and cleaved caspase 3 in Infigratinib/Bevacizumab (Figure 3, *p* < 0.05) indicated that Bevacizumab augments anti-proliferative and apoptotic activity of Infigratinib. Similar data were obtained when HCC26-0808A tumors were analyzed Appendix A).

### 2.2. Infigratinib Acts Synergistically with Bevacizumab to Inhibit Tumor Growth, Cell Invasion, Hypoxia, and LYVE-1^+^ Peritumoral Lymphatic Vessels in an HCC13-0109 Orthotopic Model

Previous studies showed that anti-VEGF agents could cause cancer cells to be more invasive [37,38]. An Infigratinib/Bevacizumab-sensitive HCC13-0109 model was selected to examine the effect of FGFR inhibition by Infigratinib, Sorafenib, Bevacizumab, and Bevacizumab plus Infigratinib on tumor growth, invasion, metastasis, and hypoxia. Due to the rapid tumor growth in this model, the treatments were carried out when the tumor size reached approximately 100–150 mm^3^ and continued for 28 days. Gross morphology examination of surgically removed liver revealed that vehicle-treated animals exhibited large tumors that spread around the liver (Figure 4a). Sorafenib and Bevacizumab reduced tumor growth but did not reduce tumor spreading. In this orthotopic model, the anti-tumor effect of Infigratinib was superior to that of Sorafenib or Bevacizumab as determined by tumor volume (Figure 4). The result of pimonidazole staining showed little to no hypoxia when tumors were treated with Infigratinib and the combination treatment. However, tumors from the Sorafenib and Bevacizumab treatment groups displayed distinct regions with intense hypoxia (Figure 4b). Compared with the control group, tumors from Infigratinib and combination treatment groups showed smooth edges and no signs of invasion. In contrast, the Sorafenib and Bevacizumab treatment group displayed uneven edges and invaded into normal liver tissue (Figure 4b). Our data show that unlike Sorafenib and Bevacizumab, which induced tumor cell invasion, micrometastasis, and significantly increased tumor hypoxia, Infigratinib and Infigratinib/Bevacizumab significantly inhibited tumor growth, tumor cell invasion, and tumor hypoxia.

It has been reported that lymphatic vessel density correlated with lymphovascular invasion and lymph node metastasis [40], as well as the invasion of tumor cells into preexisting peritumoral lymphatic vessels, conferred a significantly worse OS [41]. To determine if Infigratinib/Bevacizumab also inhibits lymphangiogenesis, LYVE-1^+^ peritumoral lymphatic vessel density was examined. Compared to the wide open peritumoral lymphatics, most of the intratumoral lymphatic vessels were small, flattened, and more inconspicuous owing to high interstitial pressure in tumors [42,43,44,45]. We also found that VEGF stimulates lymphangiogenesis in HCC13-0109 tumors, which was inhibited by Bevacizumab (Figure 4b). While Sorafenib increased the number and size of lymphatic vessel density, Infigratinib significantly reduced LYVE-1^+^ peritumoral lymphatic vessel density and area by approximately 80% and 40%, respectively. Notably, the combination of Infigratinib with Bevacizumab resulted in a significantly greater inhibition of lymphangiogenesis than a single agent. Thus, both Infigratinib and Bevacizumab synergistically suppressed peritumoral lymphangiogenesis and amplified antilymphatic activity.

### 2.3. Infigratinib/Bevacizumab Prolongs the Survival of Mice Bearing HCC Orthotopic Tumors

Since Infigratinib and Bevacizumab exerted a synergistic inhibitory effect on tumor growth, invasion, and lung metastasis, we hypothesized that Infigratinib/Bevacizumab would improve the survival rate when given to mice bearing FGFR-driven HCC tumors. A Kaplan–Meier survival analysis from 10 tumor-bearing mice showed that HCC13-0109 mice models treated with vehicle, Infigratinib, and Bevacizumab were moribund on days 50, 56, and 66, respectively (Figure 5a). Infigratinib revealed significant improvement for the mice survival rate (*p* < 0.05, log-rank test) but not with Bevacizumab. For the HCC06-0606 model, mice in the vehicle, Infigratinib, and Bevacizumab groups were moribund on days 48, 60, and 90, respectively (Figure 5b). Mice in the combination group were moribund on day 120 (HCC13-0109), and 90% of the Infigratinib/Bevacizumab-treated mice were moribund on day 130 (HCC06-0606) (Figure 5, *p* < 0.001, log-rank test). Both cell lines demonstrated greater improvement of mice OS with Infigratinib/Bevacizumab compared to Infigratinib alone. 

## 3. Discussion

HCC represents the most common primary malignant tumor of the liver and is highly dependent on angiogenesis for growth and metastasis. The complications in treating HCC are due to the various diverse molecular alterations present in individual tumors and on the abnormal activation of multiple signaling pathways that control cell proliferation, apoptosis, and angiogenesis. Considerable crosstalk and redundancy in signaling pathways are also observed in HCC. Hence, targeting single signaling pathways in HCC may have very limited efficacy, leading to a higher chance of therapy resistance. Even though blocking the angiogenesis process has been shown to prevent tumor progression in the short term, eventual progression of diseases in the presence of angiogenic therapies has been observed in clinical studies [46,47,48]. Therefore, resistance to treatment remains the major challenge for anti-angiogenic cancer therapy.

Since overexpression of FGFRs contributes to advanced HCC tumorigenesis, metastasis, and poor prognosis [13,14,15], and is postulated to be upregulated as a mechanism of resistance to anti-VEGF treatment, combination therapies designed to block FGFR/VEFGR signaling cascades may represent an alternative treatment for HCC patients. This multitarget approach can be accomplished by combining selective agents or agents that interfere with various targets [49]. In the current study, we demonstrated that the combination of Bevacizumab and Infigratinib significantly inhibited tumor growth in high FGFR-expressing HCC and Sorafenib-resistant xenografts. Greater anti-tumor activity seen in Infigratinib/Bevacizumab-treated mice can plausibly be explained by its ability to inhibit both the FGFR and VEGF signaling pathways, as well as its downstream target implicated in HCC development, metastasis, and resistance to VEGFR modulating agents [29,30,50]. In high FGFR-expressing HCC models, Infigratinib/Bevacizumab are shown to have an antitumor effect and alter the tumor microenvironment. Reduction in HIF-1α and expression of angiogenic factors (FGF2, PDGF-AA, CYR61, and VEGF) and subsequently intratumor hypoxia is due to the ability of Infigratinib/Bevacizumab to increase tumor oxygen supply via Infigratinib-induced vascular normalization of tumor blood vessels. A further decrease in tumor cell proliferation and greater induction of apoptosis without significant alteration of vascular normalization suggests minimal hypoxia and substantial antitumor efficacy. The target profile of Infigratinib/Bevacizumab is likely to confer antitumor activity in HCC tumor models that express high levels of FGFRs, particularly FGFR-2 and 3, which significantly inhibit tumor progression and invasion to prolong the survival of mice. 

The present study with Infigratinib/Bevacizumab revealed that Infigratinib-treated tumors have higher vascular densities for the sufficient supply of oxygen to tumors [51]. Partial pruning of blood vessels was performed by combining Infigratinib with Bevacizumab, which did not provoke the angiogenic rescue program nor increase tumor hypoxia. Consistent with this observation, some studies suggest vasculature pruning might improve tumor perfusion and oxygenation [52,53,54]. Our data supports the notion that tumor growth rates correlate with tumor vascular density and tumor angiogenesis under a highly regulated manner. 

The mechanistic study demonstrates a reduction in the levels of positive cell cycle regulators, including p-Rb, p-Cdc2, cyclin B1, p-p70S6K/4EBP1, Cdc25C, and survivin concomitant with increasing the levels of p27. This suggests the inhibition of cell cycle progression by Infigratinib/Bevacizumab. The apoptotic activity may be caused by the potent inhibition of MEK/ERK [55,56] and p70S6K [57] pathways. Emerging evidence has suggested that the upregulation of FGFR pathway activation may serve as an important resistance mechanism response to therapeutic pressure with the use of anti-VEGF therapy [20,58]. Thus, an upfront dual inhibition of VEGFR and FGFR, or the introduction of an FGFR inhibition after progression on a VEGF pathway inhibitor [59], may potentially result in greater clinical benefits compared to an inhibition of the VEGF pathway alone. It remains to be investigated whether the inhibition of FGFR/VEGFR by Infigratinib/Bevacizumab would lead to hyperphosphatemia and soft-tissue mineralization [60]. 

The present study showed that the synergistic inhibition of FGFR and VEGFR by combining Infigratinib/Bevacizumab resulted in sustained and significant antitumor inhibition of cancer cell progression and invasion, and improved the survival of mice. Our data support the concept that vessel normalization can have a substantial benefit, mainly by reducing metastasis, and represents a promising strategy in the treatment of FGFR-driven HCC. Targeting distinct but functionally linked cancer cell pathways at a molecular level could be a promising approach in HCC treatment. In light of the positive data from preclinical studies, a clinical trial is needed to determine if Infigratinib/Bevacizumab can provide therapeutic benefits for a subpopulation of HCC patients with FGFR-dependent tumors.

## 4. Materials and Methods 

### 4.1. Reagents

Antibodies against AKT (#9272), FGFR-3 (#4574), FGFR-4 (#8562), p70S6K (#9202), Survivin (#2803), Cdk2 (#2546), Cdc25C (#4688), Cyclin B1 (#4138), cleaved caspase 3 (#9661), cleaved PARP (#5625), Cdc2 (#9112), Rb (#9313), S6R (#2217), 4EBP1 (#9452) and α-tubulin (#2144) were obtained from Cell Signaling Technology, Beverly, MA, USA. Phosphorylation-specific antibodies against AKT Ser473 (#9271), FRS2-α Tyr439 (#3861), p70S6K Thr421/424 (#9204), S6R Ser235./236 (#2211), 4EBP1 Thr70 (#9455), Histone 3 Ser10 (#9701), Cdc2 Tyr15 (#9111), p-mTOR (#2971), p-Rb (#9308) and ERK1/2 Thr202/Tyr204 (#4370) were also obtained from Cell Signaling Technology, Beverly, MA, USA. The antibodies against FGFR-2 (sc-122), ERK1/2 (sc-94), FRS2-α (sc-17841), E2F1 (sc-251) and p27 (sc-528) were obtained from Santa Cruz Biotechnology Inc, Santa Cruz, CA, USA. The antibody LYVE-1 (AB-2988) was obtained from EMD Millipore, Billerica, MA, USA. The anti-mouse CD31 (#2502) antibody was obtained from BioLegend, San Diego, CA, USA. Infigratinib and Sorafenib were purchased from Selleck Chemicals, Houston, TX, USA. Bevacizumab (Avastin) was from Genentech, Inc., South San Francisco, CA, USA.

### 4.2. Xenograft Models

This study received ethics board approval at the SingHealth and National Cancer Centre Singapore. All animals received humane care according to the criteria outlined in the “Guide for the Care and Use of Laboratory Animals” prepared by the National Academy of Sciences and published by the National Institute of Health (NIH publication 86–23 revised 2011) [61]. HCC PDX xenograft lines were used to establish tumors in male C.B-17 SCID mice aged 9–10 weeks, weighing 23–25 g (InVivos Pte. Ltd., Singapore) [23,32]. Mice were provided with sterilized food and water ad libitum and housed in negative pressure isolators set at 23 °C and 43% humidity with 12-h light/dark cycles. 

For ectopic models, nine high (HCC06-0606, HCC26-0808A, HCC13-0109, HCC01-0909, HCC17-0211, HCC21-0208, HCC2006, HCC25-0705A, and HCC13-0212) FGFR-expressing, one Sorafenib-resistant HCC06-0606Sor46 and one low FGFR2/3-expressing HCC07-0409 models were used to study the efficacy of Infigratinib/Bevacizumab. FGFR1-3 expression levels of these lines are shown in Table 1 blow. “High” and “low” indicate an expression level above 1.5-fold and below 0.3-fold relative to normal level, respectively.

Orthotopic HCC models using HCC13-0109 and HCC06-0606 PDX models were created as previously described [39]. For the tumor invasion and metastasis study, mice bearing HCC13-0109 tumors were treated once daily with vehicle or 20 mg/kg Infigratinib, or once weekly with 5 mg/kg Bevacizumab or 20 mg/kg Sorafenib for 52 days. For the survival study, mice bearing HCC13-0109 or HCC06-0606 tumors were treated once daily with vehicle or 20 mg/kg Infigratinib for 28 days. Animals were randomized into groups of 10 mice each when the tumors reached the size of approximately 100–150 mm^3^. Body weight and OS were monitored daily. Tumor-bearing mice were sacrificed when they became moribund.

### 4.3. Development of Sorafenib-Resistant HCC Model

Mice bearing HCC06-0606 tumors were chronically treated with 10 mg/kg Sorafenib once daily. After the initial response to Sorafenib, HCC06-0606 tumors gradually acquired resistance, leading to further tumor growth. Sorafenib-treated tumors were harvested for serial transplantation after reaching 1500 mm^3^ in size. Mice bearing Sorafenib-resistant HCC06-0606 tumors were treated again with Sorafenib and harvested for implantation when they reached 1500 mm^3^. The whole cycle was repeated until Sorafenib had almost no impact on tumor growth in these resistant tumor models. The drug-resistant model, HCC06-0606Sora46 (Sorafenib-resistant passage 46), was used to study the efficacy of Infigratinib/Bevacizumab in vivo.

Nine high (HCC06-0606, HCC26-0808A, HCC13-0109, HCC01-0909, HCC17-0211, HCC21-0208, HCC2006, HCC25-0705A, and HCC13-0212) FGFR-expressing, one Sorefenib-resistant HCC06-0606Sor46, and one low FGFR2/3-expressing HCC07-0409 models were used to study the efficacy of Infigratinib/Bevacizumab.

### 4.4. Drug Treatment and Data Collection

To investigate the antitumor effects of Infigratinib/Bevacizumab, mice bearing tumor xenografts were treated as follows: (1) IP injection with 200 μL saline (vehicle); (2) 20 mg/kg Infigratinib once daily; (3) IP injection with 5 mg/kg Bevacizumab once a week (Avastin; an anti-VEGF monoclonal antibody); and (4) combined oral Infigratinib and injected Bevacizumab. Bi-dimensional measurements were performed once every 2–3 days. Tumor volumes were calculated based on the following formula: Tumor volume = [(Length) × (Width^2^) × (π/6)] and plotted as the means ± SE for each treatment group vs. time. Body and tumor weights were recorded at the time of sacrifice. The tumors were harvested 2 h after the last treatment and stored at −80 °C for later biochemical analysis. 

The efficacy of Infigratinib/Bevacizumab combination was determined by the T/C ratio, where T and C were the median weight of drug-treated and vehicle-treated tumors, respectively, at the end of treatment as previously described [23]. T/C ratios of <0.42 are considered active (Drug Evaluation Branch of the Division of Cancer Treatment, National Cancer Institute).

### 4.5. Vessel Perfusion Study

Each mouse bearing tumor xenografts (vehicle- or drug-treated) was intravenously injected with 100 mg of biotinylated Lycopersicon Esculentum (Tomato) Lectin (VectorLabs #B-1175) prepared in 100 μL of 0.9% NaCl, as previously described [23]. The tumors were harvested 10 min after lectin perfusion and fixed in 10% formalin for paraffin embedding before obtaining 5-μm sections. 

To determine the extent of hypoxia in tumor tissues, mice bearing tumors (vehicle- and drug-treated) were intraperitoneally injected with 60 mg/kg pimonidazole hydrochloride 1 h before tumor harvest.

To visualize productive microvessels, immunohistochemistry was performed using the streptavidin-biotin peroxidase complex method, according to the manufacturer’s instructions (Lab Vision Corporation, Fremont, CA, USA). Hypoxic regions of tumors were identified by staining the sections with Hypoxyprobe plus Kit HP2 according to the manufacturer’s instructions (HypoxyProbe Inc., Burlington, MA, USA).

### 4.6. Immunohistochemistry

Tumor tissues were fixed in 10% formalin in PBS at room temperature for 24 h and subsequently embedded in paraffin. Sections (5 µm) were immunostained with CD31 and p-Histone H3 Ser10, and cleaved PARP antibodies were used to assess microvessel density, cell proliferation, and apoptosis, respectively [23]. Slides were then counterstained with hematoxylin, and then dehydrated and mounted. The number of p-Histone H3 Ser10 and cleaved PARP-positive cells (among at least 500 cells per region) were counted and expressed as a number of positive cells per 1000 cells. For the quantification of mean microvessel density, 5 random fields at a magnification of ×100 were selected for each section. The number of CD31-positive of blood vessels per field was counted and expressed ± SE. All images were taken on an Olympus BX60 microscope (Olympus, Tokyo, Japan).

### 4.7. Western Blot Analysis

To determine the changes in indicated proteins, independent tumors from vehicle- and drug-treated mice were homogenized separately in a buffer containing 50 mM Tris-HCl pH 7.4, 150 mM NaCl, 0.5% NP-40, 1 mM EDTA, and 25 mM NaF, supplemented with protease inhibitors and 10 mM Na3VO4. Next, 80 micrograms of protein per sample were resolved using SDS-PAGE and transferred to a PVDF membrane [23]. Blots were incubated with indicated primary antibodies and 1:7500 horseradish peroxidase-conjugated secondary antibodies. All primary antibodies were then visualized with a chemiluminescent detection system (Advansta Inc., San Jose, CA, USA). For quantification analysis, the total density of the band corresponding to protein blotting with the indicated antibody was calculated, normalized to α-tubulin, and expressed as the fold change relative to the control (the expression level in the vehicle-treated sample). A value greater (or less) than 1 indicated that the expression level of the protein of interest was greater (or less) than that in the control group.

### 4.8. Study of Angiogenic Rescue Program

Vehicle- and drug-treated tumors were collected at indicated times and frozen at −80 °C. RNA extractions were performed according to the Qiagen RNeasy protocol. First strand cDNA synthesis was performed using the cDNA synthesis kit (Life Technology Holding Pte Ltd., Singapore). The mRNA levels of PDGF-AA, VEGF, bFGF, HIF1-α, CYR61, TGF-β1, HGF and 18S were determined by reverse transcription polymerase chain reaction (RT-PCR) with the following primers:

**Table 2 ijms-21-09405-t002:** Primers used for RT-PCR.

	Forward (5′–3′)	Reverse (5′–3′)
GAPDH	TCTCCTCTGACTTCAACAGCGACAC	TGTTGCTGTAGCCAAATTCGTTGTC
PDGF-AA	CACGGGGTCCATGCCACTAAGCAT	ATCCGGATTCAGGCTTGTGGTCGC
VEGF	CGAAGTGGTGAAGTTCATGGATG	TTCTGTATCAGTCTTTCCTGGTGAG
bFGF	TACAACTTCAAGCAGAAGAG	CGACTCTTAGCAGACATTGG
HIF-1α	ACAGCAGCCAGACGATCATGC	ACCACGTACTGCTGGCAAAGC
CYR61	TAAGGTCTGCGCCAAGCAGCTCAA	CGGCGCCATCAATACATGTGCACT
TGF-β1	CAGAAATACAGCAACAATTCCTGG	TTGCAGTGTGTTATCCGTGCTGTC
HGF	GATTCTTTCACCCAGGCATC	TTTCCTTTGTCCCTCTGCAT

Equal amount of cDNA sample derived from 100 ng of total RNA was used for each reaction and PCR was performed using the following conditions: 1 cycle at 94 °C for 2 min; 36 cycles at 94 °C for 30 s, 56 to 60 °C (dependent on the primers) for 1 min, and 72 °C for 2 min; and 72 °C for 7 min.

qRT-PCR was performed using TaqMan Real-Time PCR Master Mix (Thermo Fisher, Waltham, MA, USA) according to manufacturer’s protocol and reactions were carried out on the ViiA7 Real-time PCR system (Applied Biosystems, Foster City, CA, USA). To standardize the RNA level, we quantified the expression of GAPDH mRNA in each sample and then divided the levels of expressed bFGF, PDGF-AA, VEGF, CYR61, TGF-β1, HGF and HIF-1α mRNA by that of GAPDH.

### 4.9. Study of the Liver, Kidney and Hematological Injury-Related Parameters during the In Vivo Administration of Infigratinib/Bevacizumab

Serum derived from mice treated with vehicle, Infigratinib, Bevacizumab, and Infigratinib plus Bevacizumab for 21 days were collected, and the levels ALT, ALP, AST and creatinine were determined using the Preventive Care Profile Plus (Abaxis, Inc., Union City, CA, USA) according to the manufacturer’s instructions.

### 4.10. Statistical Analysis

GraphPad Prism 6 (GraphPad Software Inc., San Diego, CA, USA) was used for statistical analysis. For two-group comparisons, a Student’s *t*-test was used for multiple group comparisons. Further, one-way ANOVA with a post hoc Tukey analysis was used. Differences in the indicated protein levels, the tumor weight at sacrifice, the p-Histone H3 Ser10 index, the mean microvessel density, and cleaved PARP-positive cells were compared. *p* < 0.05 was considered statistically significant using an error rate of α = 0.05. The mouse survival curve was generated using the Kaplan–Meier analysis. Further, the log-rank test was used for survival analysis. Sample sizes were chosen based on our prior experience and power calculation of 85%.

## Figures and Tables

**Figure 1 ijms-21-09405-f001:**
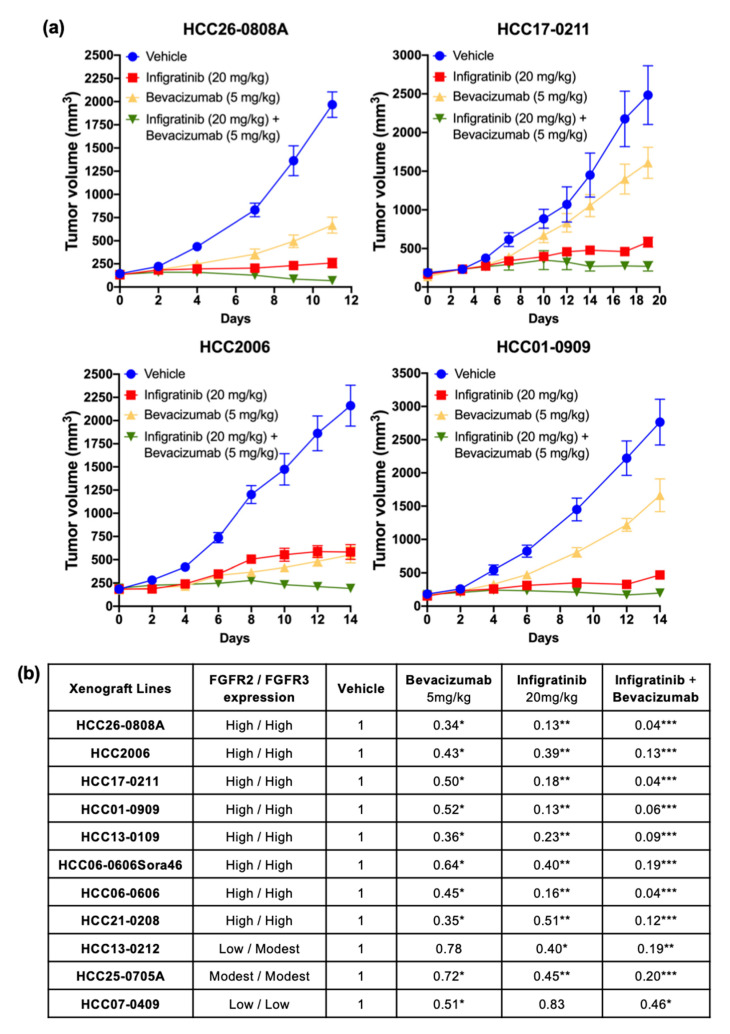
Infigratinib/Bevacizumab demonstrates potent antitumor activity in hepatocellular carcinoma (HCC) models. Tumors were implanted subcutaneously into severe combined immunodeficient (SCID) mice as described [23]. Mice (*n* = 10 per group) were subsequently treated as follows: (1) IP injection with 200 μL saline (vehicle/control), (2) 20 mg/kg Infigratinib administered orally, (3) IP injection with 5 mg/kg Bevacizumab, and (4) combined Infigratinib and Bevacizumab for indicated days. Tumor volumes were calculated and the mean tumor volumes ± SEs for 4 representative models are shown (**a**). The efficacy of Infigratinib/Bevacizumab combination was determined using the T/C ratio, where T and C are the median weights of drug- and vehicle-treated tumors at the end of treatment, respectively. The T/C ratios are shown (**b**). * *p* < 0.05; ** *p* ≤ 0.01; *** *p* ≤ 0.001 (One-way ANOVA with post hoc Tukey analysis).

**Figure 2 ijms-21-09405-f002:**
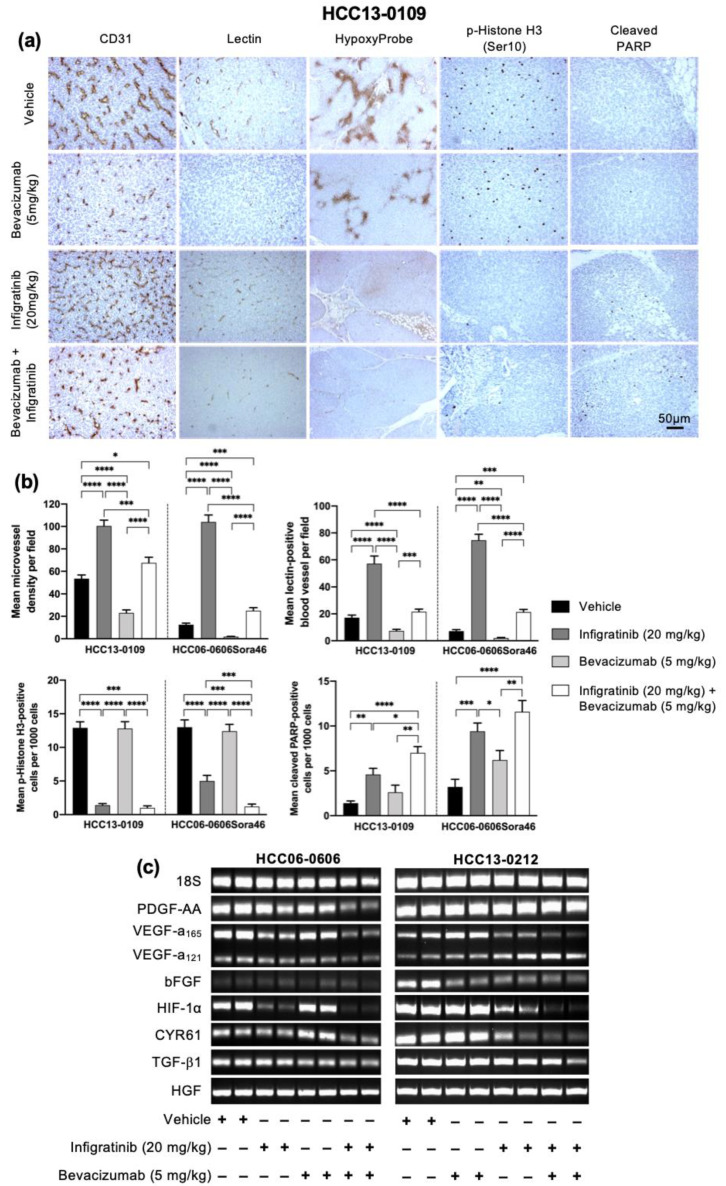
Effects of Infigratinib/Bevacizumab on angiogenesis, vessel normalization, tumor hypoxia, cell proliferation, apoptosis, and expression of angiogenic factors in HCC models. Mice bearing HCC13-0109 xenografts were treated with vehicle, Infigratinib, Bevacizumab, or Infigratinib/Bevacizumab for 14 days, as described in Figure 1. Vehicle- and drug-treated mice were perfused with biotinylated lectin and injected with pimonidazole hydrochloride as described [23]. Tumors collected 2 h after the last treatments were processed for immunohistochemistry. Representative pictures of blood vessels stained with anti-CD31, proliferative cells stained with anti-p-histone H3 Ser10, cell death stained with anti-cleaved-PARP, functional blood vessels stained with lectin and tumor hypoxia stained with HypoxyProbe antibodies are shown (**a**). Bar: 50 μm. The number of p-Histone H3 Ser10 and cleaved PARP-positive cells, among at least 500 cells counted per region was determined and plotted as the mean number of positive cells per 1000 cells ± SE. The mean lectin-positive blood vessels and microvessel density ± SE from five random fields at a magnification of ×100 was also quantified. The difference in staining-positive cells were compared using Student’s *t*-test (**b**). RNA extractions were performed according to the Qiagen RNeasy protocol. The levels of vascular endothelial growth factor (VEGF), PDGF-AA, bFGF, CYR61, TGF-β1, HIF-1α and HGF mRNA were determined using the ViiA7 Real-time PCR system. Representative ethidium bromide-stained gels are shown (**c**). The primers used for RT-PCR are listed in Table 2 (Materials and Methods). Experiments were repeated twice with similar results. * *p* < 0.05; ** *p* ≤ 0.01; *** *p* ≤ 0.001; **** *p* ≤ 0.0001.

**Figure 3 ijms-21-09405-f003:**
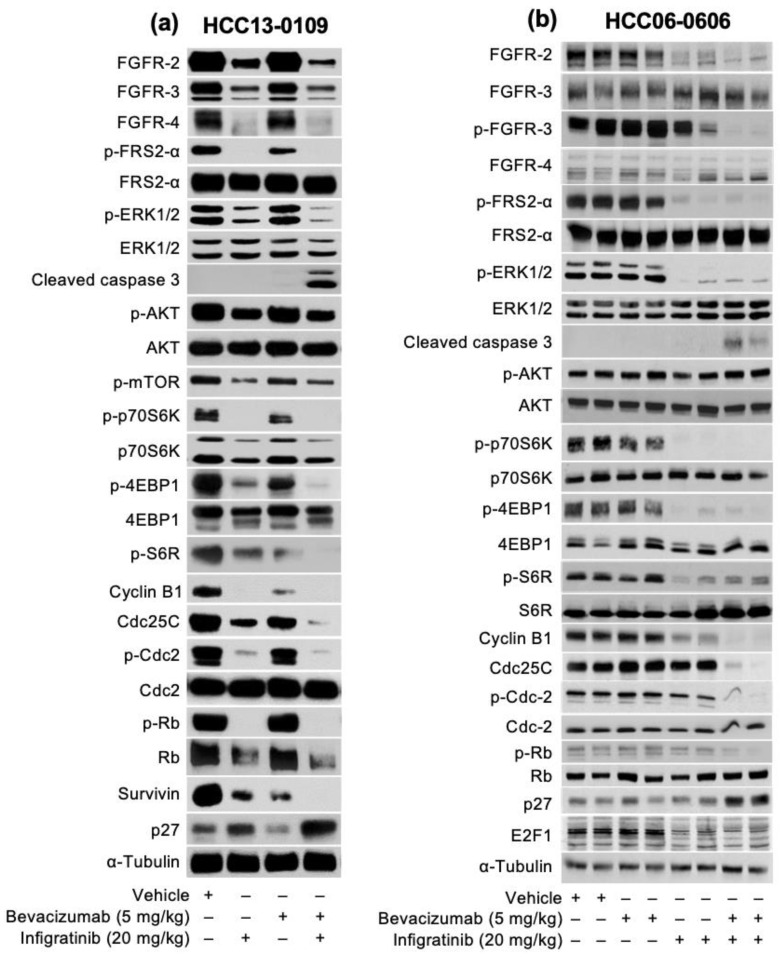
Effects of Infigratinib/Bevacizumab on expression of the fibroblast growth factor receptor (FGFR) signaling pathway and proteins involved in cell cycle, proliferation, and apoptosis in HCC13-0109 and HCC06-0606 models. Mice bearing indicated xenografts were treated as described in Figure 1 for 5 days. Tumors were collected 2 h after the last treatments and tumor lysates subjected to Western blot analysis, as previously described [23]. Representative blots for HCC13-0109 (**a**) and HCC06-0606 (**b**) are shown.

**Figure 4 ijms-21-09405-f004:**
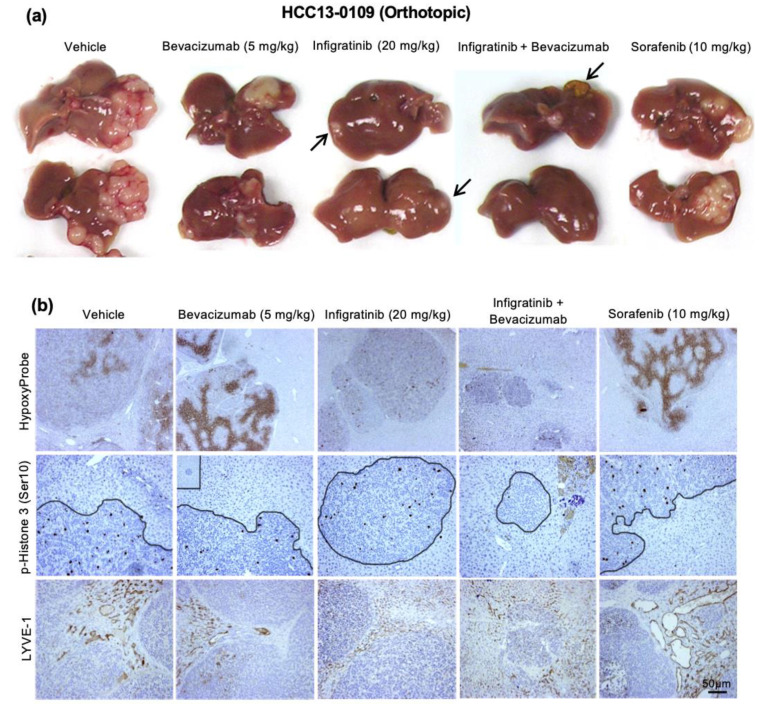
Effects of Infigratinib, Sorafenib, Bevacizumab, and Infigratinib/Bevacizumab on tumor growth, tumor hypoxia, tumor cell invasion, and peritumoral lymphatic vessel density in the HCC13-0109 model. The HCC13-0109 orthotopic model was generated as previously described [39]. Mice bearing tumor xenografts were treated with vehicle, 20 mg/kg Infigratinib daily, 10 mg/kg Sorafenib daily, 5 mg/kg Bevacizumab weekly, or 20 mg/kg Infigratinib daily plus 5 mg/kg Bevacizumab weekly for 28 days. Each treatment arm involved 10 independent tumor-bearing mice. Treatments were started when the tumors reached approximately 100–150 mm^3^. Representative pictures of primary orthotopic tumors (**a**), the extent of tumor hypoxia, tumor cell invasion, and proliferation, as well as LYVE-1^+^ peritumoral lymphatic vessel density in vehicle- and drug-treated tumors are shown (**b**). Arrows indicate tumor residues. Bars: 50 μm. Sorafenib and Bevacizumab induced tumor cell invasion, micrometastasis, and significantly increased tumor hypoxia. Infigratinib significantly inhibited tumor growth and tumor cell invasion, and reduced LYVE-1^+^ peritumoral lymphatic vessel density and area (*p* < 0.05). Combination of Infigratinib with Bevacizumab resulted in a significantly greater inhibition of tumor growth and lymphangiogenesis than Infigratinib alone (*p* < 0.05). Experiments were repeated twice with similar results.

**Figure 5 ijms-21-09405-f005:**
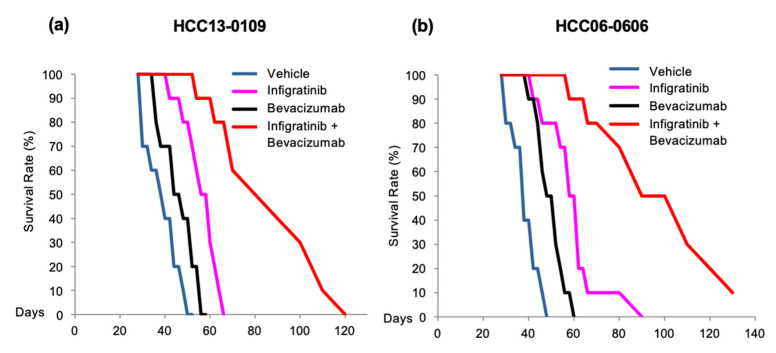
Effects of Infigratinib/Bevacizumab on the survival of mice bearing HCC13-0109 and HCC06-0606 tumors. HCC13-0109 and HCC06-0606 orthotopic models were established as previously described [39]. Mice bearing tumors were treated for 28 days, as described in the Materials and Methods. Each treatment group consisted of 10 mice. Treatments were initiated when the tumors reached sizes of approximately 100–150 mm^3^. Kaplan–Meier survival analysis for HCC13-0109 (**a**) and HCC06-0606 (**b**) models are shown. Infigratinib/Bevacizumab significantly improved the OS of mice bearing HCC13-0109 (Log-rank test, *p* = 0.0004124) or HCC06-0606 (Log-rank test, *p* = 0.0009231) compared to Infigratinib treatment alone.

**Table 1 ijms-21-09405-t001:** FGFR1–3 expression in PDX lines used in this study.

PDX Line	FGFR1	FGFR2	FGFR3
HCC13-0109	Low	High	High
HCC01-0909	Low	High	High
HCC06-0606	Low	High	High
HCC21-0208	Low	High	High
HCC26-0808A	Low	High	High
HCC17-0211	Low	High	High
HCC2006	Low	High	Modest
HCC13-0212	Low	Low	Modest
HCC25-0705A	Low	Modest	Modest
HCC06-0606Sor64	Low	High	High
HCC07-0409	Low	Low	Low

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
