# Peer review of "Bevacizumab Augments the Antitumor Efficacy of Infigratinib in Hepatocellular Carcinoma"

_ijms, 2020, doi:10.3390/ijms21249405_

Round 1
Reviewer 1 Report
Authors revised the manuscript well according to reviewers’ comments, I have no further comments.
Author Response
We thank Reviewer 1 for the positive comments.
Reviewer 2 Report
The authors present a clinically relevant problem. In the past years FGFR inhibition gained interest for the treatment of different cancers, and the recent proteomics analysis of HCC pointed out the FGF pathway amplification as a promising therapeutic target. Combination therapies with GFGR inhibitors are still novel, therefore the topic is interesting and highly relevant. Please see my detailed comments below.
Minor comments:
- Introduction is informative and well written.
- Figure 1B. Please indicate which model cell lines have high FGFR expression. To make is easier for the reader to follow, please indicate the four cell liens used for the diagrams in Figure 1A. or list the four cell lines on the top of the table.
- Missing reference at line 129: ‘Since FGFR pathway regulates a number of cellular processes including cell proliferation, 129 migration, and neo-angiogenesis [ref],’
- Figure 2. Quantitative results are necessary to present Bevacizumab’s, Infigratinib’s and the combination therapy’s effect on the capillary density, hypoxia, cell proliferation and apoptosis. Staining can be assessed by semi-quantitative methods by scanning the whole slide, randomly selecting 5-10 areas and quantifying cytoplasmic/nuclear/membraneous positivity against the number of all cells or negative cells (Aperio program or Fiji, both are free).
- Figure 5. Pleas indicate the number of mice used for determining OS
- Is there a relation between the hypoxia pathway and the FGF pathway in the endothelial cells or the hepatocytes?
Author Response
Please see the attachment

This manuscript is a resubmission of an earlier submission. The following is a list of the peer review reports and author responses from that submission.
Round 1
Reviewer 1 Report
The authors investigated whether bevacizumab and infigratinib show synergistic inhibitory effect using several in vivo models of hepatocellular carcinoma (HCC). It was suggested that combination of infigratinib with bevacizumab may possess therapeutic potential for HCC including sorafenib-resistant HCC. However, the following points should be addressed to enhance the manuscript.
- To conclude that the combination of infigratinib with bevacizumab may provide therapeutic benefits for a subpopulation of HCC patients with FGFR-dependent tumors, antitumor activity of the combination treatment should be also evaluated in the HCC models with low expression of FGFR.
- Numerous signals have been evaluated regarding the mechanisms of actions of antitumor activity of infigratinib and bevacizumab; however, the results are difficult for readers to understand. Moreover, the scientific meaning of the changes in expression levels of those signals is not logically described in the results section.
- The manuscript described “Further, 102 clinical signs of toxicity indicated that the level of toxicity of Infigratinib/Bevacizumab was acceptable.”. However, the statement was not supported by the study results driven from mice models.
- The statement of “Tumor volumes were calculated and plotted as described in [25].” is recommended to be moved to the methods section and calculation of tumor volumes and T/C ratios should be described in detail.
Reviewer 2 Report
Please see the attached file.

Reviewer 3 Report
In this study, the authors aimed to achieve a better understanding of the mechanism underlying the anti-tumor effect of concurrent inhibition of VEGFR-2 and FGFR1-3. Mice bearing high- and low-FGFR tumors were treated with Infigratinib (i.e., a pan-FGFR kinase inhibitor) and/or Bevacizumab (i.e., an angiogenesis inhibitor). They found that Infigratinib combined with Bevacizumab exerted a synergistic inhibitory effect on tumor growth, invasion, and lung metastasis, and it significantly improved the overall survival of mice bearing FGFR-dependent HCC. Infigratinib/Bevacizumab promoted apoptosis, inhibited cell proliferation concomitant with upregulation of p27, and reduction in the expression of FGFR2-4, p-FRS-2, p-ERK1/2, p-p70S6K/4EBP1, Cdc25C, survivin, p-Cdc2, and p-Rb. Although data provided here by the authors could have interesting implications, there are several concerns and problems in research design and experimental data. Specific points to be considered are listed below:
- The study used nine high FGFR-expressing HCC PDX models but only one Sorefenib-resistant HCC model. Therefore, the authors should better add more low-FGFR HCC model to conform findings in this study, because different models may have different functions.
- The authors claimed that “Infigratinib did not impair VEGF-induced blood vessel formations. Other factors, such as PDGF, TGF-α, TGF- β, FGFs, and HGF, also regulated angiogenesis.” Therefore, the authors should also evaluate PDGF, TGF-α, TGF- β, FGFs, and HGF expression in this study.
- In figure 2b for mRNA expression data, the author claimed that “mRNA qRT-PCR were determined using the ViiA7 Real-time PCR system” in legends. However, the data images are looks from Traditional PCR as amplified band on a gel. In addition, 18S rRNA as an internal control but the bands are not consistent. Furthermore, there is no information about PCR and primers in Materials and Methods.
- Although the authors used two high FGFR-expressing models of HCC06-0606 and HCC13-0212 in Figure 2b, the results described only HCC06-0606 model. In addition, that the date from the HCC13-0212 model seems to be inconsistent with model of HCC06-0606. The author need to explain results from HCC13-0212.
- The data does not match the description. In Figure 3, the authors claimed that treatment of mice with Infigratinib, but not Bevacizumab, significantly decreased p-4EBP1. However, the immunoblotting band of p-4EBP1 looks stronger in Infigratinib treatment group compared to vehicle group in HCC12-0109 model.
- The order of the groups is inconsistent in different figures. Sometimes thegroup of Bevacizumab is before the group of Infigratinib, and some other data is vice versa, which makes it difficult to track this paper.
- There are nine high FGFR-expressing HCC models in this study, however different groups are selected for each stage of the experiment and figures. The author should explain why these groups are selected for the experiment before giving the data.
Round 2
Reviewer 1 Report
Although the manuscript has been improved, it is still difficult to understand the scientific meanings and clinical implications of numerous evaluated signals. Moreover, the level of toxicity of Infigratinib is of concern.
Reviewer 3 Report
The authors have carefully addressed the reviewer's comments and improved the quality of the manuscript. I have no further comments.